# Synthesis and Evaluation of Thermoresponsive Boron-Containing Poly(*N*-isopropylacrylamide) Diblock Copolymers for Self-Assembling Nanomicellar Boron Carriers

**DOI:** 10.3390/polym11010042

**Published:** 2018-12-29

**Authors:** Shuichiro Yoneoka, Ki Chul Park, Yasuhiro Nakagawa, Mitsuhiro Ebara, Takehiko Tsukahara

**Affiliations:** 1Laboratory for Advanced Nuclear Energy, Tokyo Institute of Technology, 2-12-1-N1-6, Ookayama, Meguro-ku, Tokyo 152-8550, Japan; yoneoka.s.aa@m.titech.ac.jp (S.Y.); kimurakcpark@gmail.com (K.C.P.); 2International Center for Materials Nanoarchitectonics (WPI-MANA), National Institute for Materials Science (NIMS), 1-1 Namiki, Tsukuba, Ibaraki 305-0044, Japan; nakagawa@bmw.t.u-tokyo.ac.jp (Y.N.); EBARA.Mitsuhiro@nims.go.jp (M.E.); 3Graduate School of Pure and Applied Science, University of Tsukuba, 1-1-1 Tennodai, Tsukuba, Ibaraki 305-8577, Japan; 4Graduate School of Engineering, The University of Tokyo, 7-3-1 Hongo, Bunkyo-ku, Tokyo 113-8656, Japan; 5Innovation Center of NanoMedicine, Kawasaki Institute of Industrial Promotion, 3-25-14, Tonomachi, Kawasaki-ku, Kawasaki 210-0821, Japan; 6Graduate School of Tokyo University of Science, 6-3-1 Niijuku, Katsushika-ku, Tokyo 125-8585, Japan

**Keywords:** poly(*N*-isopropylacrylamide), nanomicelle, BNCT, boronated diblock copolymer

## Abstract

Development of new boron nanocarriers has been a crucial issue to be solved for advancing boron neutron capture therapy (BNCT) as an effective radiation treatment for cancers. The present study aimed to create a novel double-thermoresponsive boron-containing diblock copolymer based on poly(*N*-isopropylacrylamide) [poly(NIPAAm)], which exhibits two-step phase transitions (morphological transitions) at the temperature region below human body temperature. The boronated diblock copolymer considerably concentrates boron atoms into the water-dispersible (i.e., intravenous-administration possible) nanomicelles self-assembled by the first phase transition, and furthermore the properly controlled size and hydrophobicity of the second phase-transitioned nanoparticles are expected to make a significant contribution to the selective delivery and long-term retention of boron atoms into tumor tissues. Here we present the detailed synthesis of the strategic NIPAAm-based diblock copolymer with 3-acrylamidophenylboronic acid (PBA), i.e., poly(NIPAAm-*block*-NIPAAm-*co*-PBA), through a reversible addition-fragmentation chain transfer polymerization. Furthermore, the stepwise phase transition behavior of the obtained boronic-acid diblock copolymers was characterized in detail by temperature-variable ^1^H and ^11^B-nuclear magnetic resonance spectroscopy. The phase-transition-induced molecular structural changes, including the structural compositions and sizes of nanomicelles and nanoparticles, are also discussed here.

## 1. Introduction

Boron neutron capture therapy (BNCT) for cancer has been recognized as an advanced radiation treatment technique that enables targeting cancers at the cellular level. The nuclear reaction of thermal neutrons and boron-10 isotopes (^10^B, natural abundance; about 20%), which has a large thermal neutron-cross section (ca. 3,800 barn), emits alpha (α) particles and lithium atoms. Due to the short range distance (about 10 µm) of these particle beams almost comparable to the size of a single cell, only target tumor cells can be selectively killed without any damage to normal cells [1]. To achieve the selective destruction of tumor cells, therefore, boron-containing therapeutic agents capable of accumulating into tumor cells is essential in the clinical operation of BNCT. Since sodium borocaptate (BSH) and *p*-boronophenylalanine (BPA) were utilized as boron-labeled pharmaceuticals in the early clinical BNCT trials [2,3], their curative effects on various types of cancers have been proven [4,5,6,7,8,9]. However, the low-molecular boron compounds such as BSH and BPA suffer from the problem of the short retention time in tumor cells, resulting in the decrease of tumor-accumulated boron concentrations and the deterioration of the selectivity to tumor cells against normal cells. In addition, a drastic increase of boron concentrations in blood is often generated by intravenous injection, resulting in the damage of blood vessels. To overcome such disadvantages, novel boron-labeled pharmaceuticals with high boron contents and selective delivery performance are needed [1,4,10,11,12].

The enhanced permeability and retention (EPR) effect has become a crucial concept for achieving effective drug delivery into tumor cells [13,14]. Abnormal angiogenesis around tumor tissues induces gaps between the vascular endothelial cells, which leads to selective penetration of 100-nm sized materials into tumors [13,14,15]. The EPR effect utilizes the penetration of macromolecules from blood vessels into tumor tissues through the gaps [16] and the long-term retention of the macromolecules in the tumor tissues compared to small molecules. However, nowadays, it is emphasized that nanomedicine should focus on more clinically relevant tumor models in terms that the EPR effect of human cancers is extremely heterogeneous in contrast to rodent cancers [17]. Therefore, an appropriate selection of target cancers is of great importance in the nanomedicine based on the passive drug-delivery concept. More specifically, some human cancers exerting a clinically significant EPR effect, such as sarcoma [18], glioblastoma (brain tumor) [19], and head and neck cancers [20], would be promising candidates in nanomedicine-based cancer therapy including BNCT. Thus far, various kinds of boron-rich nanocarriers such as liposomes, polymeric nanomicelles, and dendric supramolecules have been synthesized, and their biochemical and pathophysiological properties have been evaluated [21,22,23,24,25,26]. However, the macromolecule-based approaches still have some shortcomings such as low boron contents per unit weight and the complicated and time-consuming synthesis operations.

A thermoresponsive polymer, poly(*N*-isopropylacrylamide) [poly(NIPAAm)], and its copolymers have been widely applied to innovative medical fields, such as new-generation drug delivery systems (DDS), particularly, nanomedicine, or regenerative medicine engineering [27,28,29]. Poly(NIPAAm) undergoes hydrophilic-hydrophobic phase transition below and above the lower critical solution temperature (LCST) of about 32 °C in aqueous solutions due to the thermally induced hydration/dehydration of the hydrophobic groups (e.g., isopropyl groups) and hydrogen-bond making/breaking between the amide groups and water molecules [30,31]. Previous studies have focused on the synthesis of poly(NIPAAm)-based materials with the ability to release the incorporated drugs selectively into tumor tissues in reaching the specific temperatures of heated tumors in hyperthermia [27,32,33,34,35]. It is also noteworthy that poly(NIPAAm)-based micelles exhibit a strong affinity with tumor tissues by hydrophobic interactions above LCST [35,36,37].

Controlled/living radical polymerization methods including reversible addition-fragmentation chain transfer (RAFT) polymerization make it possible to produce the polymers of well-defined structures. Some previous studies reported poly(NIPAAm)-based polymer-protein bioconjugates, benzoboroxole statistical copolymers and boronic-acid block copolymers synthesized by RAFT polymerization methods [38,39,40,41,42]. The unprotected boronic-acid groups are responsible for the low solubility of the boron compounds in organic solvents and the boronic-anhydride formation by dehydration, causing difficulty in the purification process. In addition, there is a concern that radical reactions are likely to be adversely affected by the high electrophilicity of boron atoms. Although there are a few synthesis examples of boronic-acid copolymers via direct polymerization without protecting boronic-acid groups [39,43,44], the adequate protection for polymerization [45,46,47] is expected to ensure the synthesis of boron-containing poly(NIPAAm)-based copolymers by preventing the adverse effect of boronic-acid groups on radical reactions and purification processes.

The development strategy of novel boron nanocarriers in this study is illustrated in Figure 1. The boron-containing diblock copolymers exhibit two-step temperature-dependent phase transition (morphological transition) due to the different hydrophilicities of the component polymer blocks. Below the phase transition temperature, the diblock copolymers are dissolved homogeneously in aqueous solutions. At the temperature above the first LCST for less hydrophilic polymer blocks, micellization occurs by the hydrophobic self-assembly of the first transitioned polymer blocks. Then, by exceeding the second LCST for more hydrophilic polymer blocks, the further morphological transformation to hydrophobic core-shell nanoparticles is caused by the contraction of the second transitioned polymer blocks. The self-assembly of the boron-containing diblock copolymers makes it realized to materially concentrate boron atoms to the nanomicelles and nanoparticles as boron carriers. Advantageously, the two-step LCST lower than human body temperature will achieve facile administration of the hydrophilic self-assembled nanomicelles by intravenous injection and the further *in-vivo* phase transition to the hydrophobic core-shell nanoparticles playing a role as boron carriers. Furthermore, the size control of the nanoparticles (about 100 nm) enables to take advantage of the EPR effect, and their hydrophobicity contributes to hydrophobic interaction with cancer tissues. Based on the strategic development concept, we aimed to synthesize novel thermoresponsive poly(NIPAAm)-based diblock copolymer with 3-acrylamidophenylboronic acid (PBA), i.e., poly(NIPAAm-*block*-NIPAAm-*co*-PBA), by using a simple RAFT polymerization method (Scheme 1). Thus far, RAFT polymerization has been applied to the synthesis of versatile block copolymers [26,33,45,46,48,49,50]. The characterization of the novel boronic-acid diblock copolymer was conducted by ^1^H and ^11^B-nuclear magnetic resonance (NMR) spectroscopy, gel permeation chromatography (GPC), optical transmittance measurements in ultraviolet-visible (UV–Vis) spectroscopy and dynamic light scattering (DLS). Here we present the synthesis procedures of poly(NIPAAm-*block*-NIPAAm-*co*-PBA) and the characterization results including the findings on the particulates produced in the stepwise phase transition.

## 2. Experimental

### 2.1. Materials

*N*-Isopropylacrylamide (NIPAAm), 2-2′azobisisobutyronitrile (AIBN) and diethanolamine were purchased from Wako Pure Chemical (Osaka, Japan). NIPAAm and AIBN were purified by recrystallizing from hexane for three times and from acetone, respectively. PBA and cumyldithiobenzoate (CDB) were available from Sigma-Aldrich (St. Louis, MO, USA), and used as received. Methanol-d_4_ (CD_3_OD; 99.96 atom %D), deuterium oxide (D_2_O; 99.96 atom %D), chloroform-d (CDCl_3_; 99.96 atom %D), and dimethyl sulfoxide-d_6_ ((CD_3_)_2_SO; 99.9 atom %D) were purchased from Acros Organics (Morris Plains, NJ, USA). All other reagents were purchased from Kanto Chemical (Tokyo, Japan) and used as received.

### 2.2. Synthesis of Boron-Containing Poly(NIPAAm)-Based Copolymer and Diblock Copolymer via RAFT Polymerization

NIPAAm copolymers with PBA, poly(NIPAAm-*co*-PBA) was synthesized as follows. Prior to RAFT copolymerization, the boronic-acid groups of PBA were protected by diethanolamine (DEA). PBA (500 mg, 2.62 mmol) was dissolved in 50 mL of the argon (Ar)-degassed tetrahydrofuran (THF). DEA (275 mg, 2.62 mmol) was gradually added into the solution, and refluxed for 6 h at 65–68 ℃ with Ar gas bubbling. The resulting precipitates were suction-filtered and dried overnight in a vacuum at 30 °C to recover protected PBA (in Scheme 1
**[1]**) as white fine powders.

The protected PBA (294 mg, 1.13 mmol), NIPAAm monomers (1.14 g, 10.0 mmol), CDB (20.4 mg, 0.0750 mmol) as RAFT reagents, and AIBN (2.43 mg, 0.0148 mmol) as initiators were added to 11.0 mL of dimethyl sulfoxide (DMSO) degassed by freeze-pump-thaw cycles three times, and reacted at 70 °C for 24 h in an Ar-filled glove box. Then, the solution was dialyzed overnight using a regenerated cellulose membrane tubing (Spectra/Pro 6, MWCO: 1 kDa, Spectrum Laboratories, Inc., Rancho Dominguez, CA, USA) in 500 mL of ethanol and the dialyzed solution was evaporated under reduced pressure. The resulting crude solid was dissolved in a few mL of tetrahydrofuran (THF) and dropped into about 500 mL of dehydrated diethyl ether (Et_2_O) to form pale pink precipitates. Then, the precipitates were suction-filtered and dried in a vacuum to recover poly(NIPAAm-*co*-PBA(protected)) (in Scheme 1
**[2]**) as pale pink powders.

For comparison, poly(NIPAAm) was also synthesized basically according to the same RAFT polymerization under the quantitative condition of NIPAAm (1.15 g, 10.1 mmol), CDB (20.4 mg, 0.0750 mmol), and AIBN (2.43 mg, 0.0148 mmol) in 15 mL of degassed DMF. The resulting poly(NIPAAm) was recovered through the same dialysis and solidification processes as mentioned above.

Removal of the protecting groups from poly(NIPAAm-*co*-PBA(protected)) was performed by the following easy hydrolysis [51]. About 100 mg of poly(NIPAAm-*co*-PBA(protected)) was added to 10 mL of distilled water. The resulting mixture was cooled down to 1 °C and reacted with 10 mL of 0.2 M hydrochloric acid (HCl) with stirring for 2 h. Then, the solution was neutralized by sodium hydroxide (NaOH) aqueous solution and heated up to 35 °C to form pale pink precipitates. After suction-filtering and drying overnight at 30 °C, deprotected poly(NIPAAm-*co*-PBA) was recovered as pale pink powders.

The diblock copolymer comprised of poly(NIPAAm) and poly(NIPAAm-*co*-PBA) blocks, i.e., poly(NIPAAm-*block*-NIPAAm-*co*-PBA) (in Scheme 1
**[3]**), was synthesized as below. NIPAAm (573 mg, 5.02 mmol), AIBN (1.23 mg, 0.00749 mmol), and poly(NIPAAm-*co*-PBA(protected)) (142 mg, 0.0351 mmol) were mixed in 7.5 mL of the dimethyl sulfoxide (DMSO) degassed by Ar bubbling for 30 min. RAFT polymerization was performed according to the same reaction conditions as poly(NIPAAm-*co*-PBA(protected)). Deprotection of PBA was conducted by the same acid treatment as mentioned above to recover poly(NIPAAm-*block*-NIPAAm-*co*-PBA) as pale pinkish white powders.

### 2.3. Characterization of Boron-Containing Poly(NIPAAm)-Based Block Polymers

The molecular structures of the synthesized poly(NIPAAm)-based polymers were investigated by ^1^H and ^11^B-NMR spectroscopy at various temperatures. ^1^H and ^11^B-NMR spectra were recorded on a JEOL JNM-ECX400P instrument (JEOL Ltd., Tokyo, Japan) at the resonance frequencies of 400 and 128 MHz, respectively. The deuterated solvents, methanol-d_4_, deuterium oxide, chloroform-d and DMSO-d_6_ were used for shimming and locking. ^1^H-NMR chemical shifts were determined against 3-(trimethylsilyl)-1-propanesulfonic acid-d_6_ sodium salt (DSS-d_6_) as an internal reference. For ^11^B-NMR, boron trifluoride diethyl ether complex (BF_3_·OEt_2_) was adopted as an internal or external reference. Only in the case of D_2_O solvent, BF_3_·O Et_2_ was used as an external reference by loading it to the interstice of an NMR double tube.

The GPC measurements were performed at 40 °C using TSKgel G3000H_HR_ columns (TOSHO, Tokyo, Japan) and a refractive index detector (RI-2031, JASCO International Co., Ltd., Tokyo, Japan) to determine the weight-average molecular weight (*M*_w_), number-average molecular weight (*M*_n_), and molar-mass dispersity (*Ð* = *M*_w_/*M*_n_) of the obtained polymers. THF was used as a mobile phase at the flow rate of 1.0 mL·min^−1^. Polystyrene with molecular weight (*M*_w_) of 2630, 5060, 10,200, 17,400, and 37,900 were used as standard samples for making calibration curves.

The LCST of the poly(NIPAAm)-based polymers was determined by measuring the temperature-dependent variation of optical transmittance of the polymer solution at 500 nm. The sample solutions were prepared by dissolving the polymers in ultrapure water (Milli-Q, Millipore, Merck KGaA, Darmstadt, Germany) at the concentration of 10 mg·mL^−1^. The optical transmittance was recorded on UV–Vis spectrophotometer (V-630, JASCO Corp. Tokyo, Japan) with a Peltier temperature controller. The temperature was controlled at the heating rate of 0.1 °C·min^−1^.

The particle sizes of nanomicelles and nanoparticles derived from the poly(NIPAAm)-based diblock copolymers by the stepwise phase transition of each polymer block at the different LCST were examined by means of DLS measurements (Delsa^TM^ Nano HC, Beckman Coulter, Inc., Brea, CA, USA). The DLS measurements were performed using the diblock polymer solutions with enough low concentrations of 2.0 or 20 μg·mL^−1^ to prevent the agglomeration and precipitation of the diblock polymers at the temperature range of 16 to 28 °C or 32 to 40 °C, respectively.

The critical micelle concentration (CMC) of the boron-containing diblock copolymer was determined from the concentration-dependent variation of optical transmittance of the polymer solutions at the fixed wavelength of 400 nm (at which the diblock copolymer has no absorbance). The optical transmittance was measured at 20 °C by using UV–Vis spectrophotometer (V-630, JASCO Co.). The scattering of irradiation light on the micelles formed above CMC is responsible for unambiguous decrease of optical transmittance. Therefore, CMC can be determined as the cross point of the two least squares regression lines of the data before and after micellization of the diblock copolymer.

## 3. Results and Discussion

Quantitative protection of boronic-acid groups of PBA with DEA was confirmed by ^1^H-NMR spectroscopy. Thus far, diols such as pinacol [52] and pinanediol [53,54] have been extensively utilized for protecting boronic-acid group, due to the effectiveness as protecting groups by the bulkiness and high stability of the resulting boronate esters. However, the high protection ability makes it difficult to deprotect the cyclic boronates. More specifically, pinanediol and pinacol [55,56,57], boronate esters were not affected by hydrolysis. Therefore, not common methods, such as a two-step procedure comprised of transesterification with DEA followed by hydrolysis [51,52], conversion to boronic acids via trifluoroborate intermediates [58], or a two-phase transesterification procedure with polystyrene-supported boronic acid [59], have been applied to the deprotection of pinacol boronates. In contrast, the deprotection of DEA boronate esters can be quantitatively achieved by common hydrolysis methods under mild conditions [51]. In the present study, therefore, DEA was employed for protection of boronic-acid groups. In the ^1^H spectrum of the product recovered after the protection process (see Appendix A), new peaks appeared at 2.8–3.9 ppm, which were not observed in the pristine PBA (see Appendix A). These new peaks were assigned as the –CH_2_– group (peaks; **6, 7**) of DEA.

Figure 2a shows the ^1^H-NMR spectra of poly(NIPAAm-*block*-NIPAAm-*co*-PBA) in CD_3_OD/DMSO-d_6_ (=1:1) mixture solvent at ambient temperature. The ^1^H-signals at around 1.0 ppm, 1.5–2.2 ppm, and 4.0 ppm were assigned to –CH_3_ group (peak; **e**), copolymer main chains (peaks; **1**, **2**, **a**, **b**), and –CH– group (peak; **d**), respectively. The multiple peak at about 7.3–7.8 ppm was assigned to the phenyl groups of PBA and CDB (peaks; **3**, **f**, **g**) (see the inset in Figure 2a). In the case of this mixture solvent, the peaks originating from the –NH groups of the polymer side chains and the –OH groups of boronic-acid moieties are not observed because of the H/D exchange between the –NH and –OH protons of the diblock copolymer and the –OD deuteron of CD_3_OD. To assign –NH and –OH groups, therefore, the ^1^H-NMR spectrum of poly(NIPAAm-*block*-NIPAAm-*co*-PBA) in DMSO-d_6_ was also measured. As seen from Figure 2b, the obtained ^1^H-NMR spectrum provided not only the similar peaks assigned to the copolymer main chain (peaks; **1**, **2**, **a**, **b**), –CH– (peak; **d**),–CH_3_ (peak; **e**), and the phenyl groups of PBA and CDB (peaks; **3**, **f**, **g**) but also new peaks in the low magnetic-field region (see the inset of Figure 2b). The new peaks at about 8.0 ppm and 9.5 ppm originate from –NH (peaks; **4**, **e**) of amide groups and –OH (peak; **5**) of boronic-acid groups, respectively. The assignment of all ^1^H-NMR peaks of poly(NIPAAm) and poly(NIPAAm-*co*-PBA) (Appendix A) were consistent with that of Figure 2, which also supports the successful synthesis of poly(NIPAAm)-based boronic-acid diblock copolymers. 

The ^11^B-NMR spectrum of poly(NIPAAm-*block*-NIPAAm-*co*-PBA) measured in DMSO-d_6_ at an ambient temperature is shown in Figure 3. The broad peak observed at around 21 ppm was attributable to the boronic-acid groups of poly(NIPAAm-*block*-NIPAAm-*co*-PBA) in terms of the larger chemical shift than protected boronic acids (e.g., unprotected and protected PBA monomer; ca. 29 and 9 ppm, respectively). The ^11^B-NMR results indicate that the RAFT polymerization of the protected boronic-acid copolymers proceeds without being interfered by the electrophilicity of boron atoms, and the protecting groups are readily removed by the acid-catalyzed hydrolysis using HCl solution. To estimate the amount of boron atoms in the diblock copolymers, the quantitative analysis of ^11^B-NMR was performed according to the following relationship.
$$\begin{array}{c} \left[\mathrm{amount}\;\mathrm{of}\;\mathrm{boron}\;\mathrm{atoms}\;\mathrm{in}\;a\;\mathrm{copolymer}\right]\\ \;=\frac{\left[{\mathrm{BF}}_{3}\;\mathrm{boron}\;\mathrm{concentration}\right]\times \left[\mathrm{DMSO}\;\mathrm{volume}\right]}{\left[\mathrm{amount}\;\mathrm{of}\;\mathrm{copolymer}\;\right]/\;\left[\mathrm{copolymer}\;\mathrm{molecular}\;\mathrm{weight}\right]}\times \left[\mathrm{peak}\;\mathrm{area}\;\mathrm{ratio}\right] \end{array}$$

The peak area ratio of polymer to BF_3_·OEt_2_ was 0.88, which was analyzed by spectral-curve fitting with a Lorentzian function. The NMR sample was composed of DMSO-d_6_ solvent (750 μL), BF_3_·OEt_2_ internal reference (7.9 × 10^−3^ M), and the diblock copolymer (41.3 mg) with the molecular weight of 2.0 × 10^4^ g·mol^−1^ (obtained from GPC data shown in Table 1). From the calculation result, it was found that the diblock copolymer contains 2.9 boron atoms per single polymer molecule. The value larger than 1.0 means a significant deviation from random copolymerization, suggesting that PBA monomers exhibit higher reactivity than NIPAAm monomers in the RAFT polymerization process. From the result of ^11^B-NMR quantitative analysis, PBA content was determined as 2.7 mol %.

The results of GPC analysis, i.e., the *M*_w_, *M*_n_, and *Ð* values of poly(NIPAAm), poly(NIPAAm-*co*-PBA), and poly(NIPAAm-*block*-NIPAAm-*co*-PBA) and those values calculated from ^1^H NMR spectra (Appendix A) are listed in Table 1, which also includes the molar ratios of reaction mixture in RAFT polymerization and theoretical *M*_w_. The chromatograms used for the analysis are shown in Appendix A. 

The *M*_w_ and *M*_n_ values of poly(NIPAAm-*block*-NIPAAm-*co*-PBA) determined by GPC analysis were estimated as 3.7 × 10^4^ and 2.0 × 10^4^, respectively. The *Ð* values of the NIPAAm-based polymers suggests that the controllability of RAFT polymerization of NIPAAm-copolymer blocks is comparable to that of NIPAAm-homopolymer blocks. All the findings derived by the ^1^H and ^11^B-NMR and GPC analyses have clearly demonstrated the successful synthesis of poly(NIPAAm)-based boronic-acid diblock copolymers.

To confirm the thermally induced phase transition behavior of poly(NIPAAm)-based polymers, the LCST of the polymers in pure water was determined by temperature-variable UV–Vis measurements. Figure 4 shows the temperature-dependent optical transmittance variation of poly(NIPAAm-*block*-NIPAAm-*co*-PBA) aqueous solutions and its temperature-differential curve. The similar plots of optical transmittance of poly(NIPAAm) and poly(NIPAAm-*co*-PBA) solutions are also shown in Appendix A. It was confirmed that the optical transmittance of the poly(NIPAAm) solution was sharply reduced at the temperature range of 29–32 °C, as is shown in Appendix A. In contrast, poly(NIPAAm-*co*-PBA) solution exhibited the less sharp variation at the lower and wider temperature range of 7–14 °C. The phenyl group of PBA makes poly(NIPAAm-*co*-PBA) less hydrophilic than poly(NIPAAm), which lowers the LCST of the copolymer compared to poly(NIPAAm). As seen from Figure 4a, the poly(NIPAAm-*block*-NIPAAm-*co*-PBA) solution showed the two-step reduction of optical transmittance arising from the individual phase transition at the first and second LCSTs, which were determined as 15.5 and 30.5 °C from Figure 4b, respectively. The first gentle reduction of optical transmittance was observed at the lower temperature region corresponding to that of poly(NIPAAm-*co*-PBA), while the higher temperature range for the second sharp reduction was almost equivalent to that of poly(NIPAAm). It is therefore concluded that the phase transitions of the diblock copolymer at the first and second LCSTs originate from the poly(NIPAAm-*co*-PBA) block and poly(NIPAAm) block, respectively.

Temperature-variable NMR spectroscopy provides a detailed information on the molecular structural changes of the diblock copolymer in the two-step phase transition in terms that each partial structure can be observed individually. Figure 5a shows the temperature-dependent ^1^H-NMR spectra of the D_2_O solution of poly(NIPAAm-*block*-NIPAAm-*co*-PBA) ranging from 4 to 40 °C. The ^1^H-NMR spectrum before phase transition provided analogous ^1^H-signals to those observed in the CD_3_OD/DMSO-d_6_ mixture solvent [1.0 ppm (–CH_3_ group), 1.5–2.0 ppm (–CH_2_CH– polymer main chain), 4.0 ppm (isopropyl –CH– (**d**)), and 7.0–8.0 ppm (phenyl group (**3**) of boronic-acid moieties)]. As an overall trend, the peak intensities of all the signals were decreased with increasing temperature, and almost disappeared above 36 °C. However, it should be noted that there is a difference in the temperature effect on the ^1^H-peak intensities between the isopropyl –CH– (**d**) and phenyl groups (**3**). The peak intensity of the phenyl group (**3**) was decreased drastically at the vicinity of 24 °C and disappeared at 32 °C, while an abrupt thermal response of the isopropyl –CH– (**d**) occurred at more than 32 °C. Figure 5b shows the temperature dependence of the relative ^1^H-peak intensities of the phenyl (**3**) and –CH_2_CH– polymer main chains (**1**, **2**, **a**, and **b**) against –CH_3_ (**c**) groups. The variation profile of [phenyl]/[–CH_3_] presented the definite reduction at the lower temperature region of 4 to 16 °C, while the ratio of [–CH_2_CH–]/[–CH_3_] was sharply decreased at the higher temperature of more than 32 °C. This trend was almost comparable with that observed in temperature-variable UV–Vis measurements.

The influence of phase transition on the chemical environment and structures of boronic-acid groups of the diblock copolymers was investigated by using temperature-variable ^11^B-NMR measurements. Figure 6a shows some typical ^11^B-NMR spectra of D_2_O solution of poly(NIPAAm-*block*-NIPAAm-*co*-PBA) at the temperature of 4–40 °C (BF_3_·OEt_2_ was used as an external reference). The broad ^11^B-signal of the boronic-acid –O–B–O– groups was observed at around 27 ppm. The variation of the ^11^B-signal was ambiguous owing to the small amount of the polymer sample, which resulted from the lower solubility in D_2_O compared with DMSO-d_6_ and the small volume of the NMR double tube. Accordingly, the ^11^B-NMR spectra were analyzed by spectral-curve fitting with a Lorentzian function to determine the chemical shifts and peak area intensities. For direct comparison, furthermore, the peak area intensities calculated were normalized as a reference of the similarly calculated peak area intensity of BF_3_·OEt_2_. As shown in Figure 6b, the ^11^B-chemical shift and normalized signal intensity of the –O–B–O– groups were almost constant independently of the temperature variation across the first and second LCSTs. Moreover, the ^11^B signals definitely remained even at 40 °C beyond the second LCST, at which the ^1^H-NMR peaks disappeared. The results indicate that thermal-induced dehydration around the diblock copolymer leads to shrinkage of the polymer chains at each LCST, whereas the chemical structure of boronic-acid groups and their hydration environment is stable regardless of morphological transformation by the two-step phase transition.

The hydrodynamic diameters of the diblock copolymer-based particulates derived by the two-step phase transition were determined by DLS measurements at the temperature region above the first LCST (>15.5 °C). The diblock copolymers were self-assembled to nanomicelles by hydrophobic interactions between the copolymer blocks above the first LCST, which maintain dispersed state in water. As seen from Figure 7, the hydrodynamic particle size was constant at 50–60 nm in the temperature range of 16 to 28 °C. It is considered that the water-dispersible nanomicelles are comprised from the inner cores of copolymer blocks and the water-soluble outer shells of NIPAAm-homopolymer blocks. In terms of the difficulty of light scattering on water-soluble outer shell, it is considered that the determined particle size of 50–60 nm would originate from the hydrophobic inner cores. The CMC determined from the concentration-dependent variation of transmittance by the scattering of irradiation light on the nanomicelles was ca. 2.8 μM (Appendix A). The quite dilute micellization concentration of the boronated diblock copolymer allows adjustable preparation of nanomicelle solution for clinical use in a wide concentration range. At the higher temperature range of 32–40 °C above the second LCST, the nanocore-micelles further transitioned to hydrophobic nanoparticles with the larger and constant diameter of about 80 nm, which exhibit aggregation tendency in water. Above the second LCST (>32 °C), the contraction of the poly(NIPAAm) blocks by further hydrophobic interactions was induced, resulting in the formation of hydrophobic core-shell nanoparticles.

The number of diblock copolymers in the core-shell nanoparticles can be calculated by using the total mass of the diblock copolymers in the aqueous solution and the molecular weight *M*_n_, the total volume of the nanoparticles formed (for example, at 40 °C), the single-nanoparticle volume derived from the obtained particle diameter, and the density of water (d_40_; 0.99222 g·cm^−3^ [60]). As the result of calculation, the core-shell nanoparticle is found to be comprised of 8.6 × 10^3^ diblock copolymer molecules per particle. Furthermore, each diblock copolymer contains 2.9 boron atoms per single polymer molecule, as was already mentioned in the quantitative analysis of ^11^B-NMR. Totally, therefore, the core-shell nanoparticle contains 2.5 × 10^4^ boron atoms per particle. Considering that BPA used in the current clinical BNCT contains one boron atom per molecule, the core-shell nanoparticle containing the far more enormous amount of boron atoms is expected to be a promising boron carrier. In addition, it was reported that doxorubicin-loaded radiolabeled liposomes (typical diameter; about 30–105 nm [61]) showed high tumor-to-normal tissue accumulation to some human cancers, such as sarcoma [18], glioblastoma [19], and head and neck cancers [20]. Therefore, the combination of the strong EPR effect of the specific cancers with the hydrophilicity, controlled size (about 80 nm), and enormous boron content of the core-shell nanoparticles allows us to expect the potential applicability of the boronated diblock copolymers to BNCT.

## 4. Conclusions

A novel thermoresponsive boron-containing diblock copolymer, i.e., poly(NIPAAm-*block*-NIPAAm-*co*-PBA), was successfully synthesized through a simple RAFT polymerization in the presence of boronic-acid protecting groups, followed by acid-catalyzed hydrolysis for the easy-deprotection of the boronate. The poly(NIPAAm)-based diblock copolymer exhibits the two-step phase transition at the first and second LCSTs of 15.5 and 30.5 °C to form hydrophilic nanocore-micelles and hydrophobic core-shell nanoparticles, respectively. The hydration environments around the boronic-acid groups are found to be maintained before and after the two-step phase transition, in sharp contrast to the significant change observed in the hydrophobic CH components such as the isopropyl groups, polymer main chains, and the phenyl groups of boronic-acid moieties. The diblock copolymers contain 2.9 boron atoms per single polymer molecule. Therefore, the self-assembly of the diblock copolymers materially concentrates boron atoms into the inner cores of the nanomicelles. The diblock copolymers provide the water-dispersible, boron-rich nanocore-micelles and furthermore the well-controlled size (about 80 nm) of hydrophobic core-shell nanoparticles, which contain enormous content of 2.5 × 10^4^ boron atom per particle. Therefore, we expect that the core-shell nanoparticles self-assembled by the novel boronated diblock copolymers have a potential utility to BNCT as a new type of boron nanocarrier based on the drug-delivery concept of the EPR effect and hydrophobic affinity to tumor tissues.

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
