# Peer review of "Synthesis and Evaluation of Thermoresponsive Boron-Containing Poly(N-isopropylacrylamide) Diblock Copolymers for Self-Assembling Nanomicellar Boron Carriers"

_polymers, 2018, doi:10.3390/polym11010042_

Round 1

Reviewer 1 Report

The manuscript by Tsukahara and co-workers present new block copolymers based in poly(N-isopropylacrylamide) and 3-acrylamidophenylboronic acid with the aim to be used in boron neutron capture therapy. Block copolymers with two NIPAM based thermoresponsive blocks have been synthesized, one NIPAM homopolymer and a second block of random NIPAM-PBA copolymer .

The polymers have been thoroughly characterized however there is no in vitro work or any relevant experiments to, at least, direct the narrative towards the actual use of these polymers for BNCT.

This manuscript could potentially be accepted for the main reason that there are not many reports on boronic acid based (block) copolymers synthesized with controlled radical polymerizations (including RAFT) other than the works by the Sumerlin group (the researchers accurately cite these works in their manuscript).

However, there are minor and major issues that must be addressed before considering acceptance in Polymers:

-Typographical errors and some text polishing is needed throughout (i.e. typos in line 221) (minor).

- For the synthetic part, diethanolamine is used as a protecting group although the most common synthetic route is the pinacol boronate ester formation widely reported in the literature. The researchers should give an explanation for their design rationale (minor).

- The authors should discuss the boron content of the block copolymer for use BNCT. Is it adequate? Is it clinically relevant? (minor).

- The narrative of the EPR effect for the passive targeting of polymers to solid tumors is rather outdated. The EPR has been observed in animal models but is not abundant in human malignancies; the phraseology and the literature on this matter must be carefully updated to keep in line with current findings (minor).

- The researchers report ca. 2 PBA monomers per polymer chain. Is there any overlapping of the PBA aromatics with the chain transfer agent in the NMR spectra (Figure 2)? If yes, this is critical given the relative low degree of polymerization (DP) values used and hence the peak assignments in the NMR spectra must include the CTA peaks too (minor).

- Similarly, to avoid renal clearance of polymers intravenously injected, high molecular weight polymers should be used. Instead, here the maximum Mn reported for the block copolymer is 17,000Da whereas for these polymer nanomedicines we are looking for Mn of at least 80-100 kDa. The researchers must provide experimental data that it is possible to obtain block copolymer batches with high degree of polymerization (DP) using their RAFT experimental protocol, with acceptable PDI (major).

- The critical micelle concentration value of the block copolymer must be reported to clearly prove the potential use of these polymers for BNCT in their assembled form upon IV injection. The pyrene assay could be easily performed with these polymers (major).

Author Response

To Reviewer 1

We deeply appreciate your courtesy to review our manuscript. Your comments are quite helpful for us to improve the quality of our manuscript. We have made our best efforts to answer your questions and to comply with your advice as much as possible. In addition, we have carefully checked out all data, body text and references again. In this process, we have found some mistakes, and corrected the mistakes appropriately (please see below for the details). We sincerely apologize for our careless mistakes. We would like to emphasize that the major context of “Results and discussion” and “Conclusions” is not changed from that of the original manuscript by this amendment. All the amendments are highlighted with green and sky blue makers in the revised manuscript. The green-highlighted amendments correspond to our response to the questions and requests from you and another reviewer, and the sky blue-highlighted ones are the amendments by our own to improve the manuscript.

We would appreciate it if you could spare your precious time for further consideration of our answer below and revised manuscript.

(Response to Query-1)

According to your advice, we checked out the text and figures throughout the manuscript, and corrected some mistakes including typographical errors. We would like you to see the parts highlighted by blue marker for the amendment. In this process, furthermore, we have noticed wrong selection of GPC data. Therefore, we revised GPC chromatograms (Figure S3) and Table 1 using the correct data. In accordance with the corrected Table 1, the explanation about dispersity (Ð) in the RAFT polymerization of poly(NIPAAm) and poly(NIPAAm-co-PBA) was revised (please see Line 278–280). In addition, the boron content of the diblock copolymer was re-calculated by using the corrected molecular weight of the diblock copolymer and changed from 2.1 to 2.9. We sincerely apologize for our careless mistakes. Furthermore, some additional literatures were cited, and all of the reference No. were appropriately changed in the revised manuscript.

(Response to Query-2)

The explanation about the use of diethanolamine (DEA) as a protecting group of boronic acid moiety was added in the revised manuscript, which is as follows.

Line 214–224:

Thus far, diols such as pinacol[52] and pinanediol[53,54] have been extensively utilized for protecting boronic-acid group, due to the effectiveness as protecting groups by the bulkiness and high stability of the resulting boronate esters. However, the high protection ability makes it difficult to deprotect the cyclic boronates. More specifically, pinanediol and pinacol[55-57] boronate esters were not affected by hydrolysis. Therefore, not common methods, such as a two-step procedure comprised of transesterification with DEA followed by hydrolysis[51,52], conversion to boronic acids via trifluoroborate intermediates[58], or a two-phase transesterification procedure with polystyrene-supported boronic acid[59], have been applied to the deprotection of pinacol boronates. In contrast, the deprotection of DEA boronate esters can be quantitatively achieved by common hydrolysis methods under mild conditions[51]. In the present study, therefore, DEA was employed for protection of boronic-acid groups.

(Response to Query-3)

We discussed the applicability of the boron-containing diblock copolymer to BNCT by calculating the total boron content of the core-shell nanoparticle as a boron carrier, as described below.

Line 368–383:

The number of diblock copolymers in the core-shell nanoparticles can be calculated by using the  total mass of the diblock copolymers in the aqueous solution and the molecular weight Mn, the total volume of the nanoparticles formed (for example, at 40 °C), the single-nanoparticle volume derived from the obtained particle diameter, and the density of water (d40; 0.99222 g cm-3[60]). As the result of calculation, the core-shell nanoparticle is found to be comprised of 8.6×103 diblock copolymer molecules per particle. Furthermore, each diblock copolymer contains 2.9 boron atoms per single polymer molecule, as was already mentioned in the quantitative analysis of 11B-NMR. Totally, therefore, the core-shell nanoparticle contains 2.5×104 boron atoms per particle. Considering that BPA used in the current clinical BNCT contains one boron atom per molecule, the core-shell nanoparticle containing the far more enormous amount of boron atoms is expected to be a promising boron carrier. In addition, it was reported that doxorubicin-loaded radiolabeled liposomes (typical diameter; about 30–105 nm[61]) showed high tumor-to-normal tissue accumulation to some human cancers, such as sarcoma[18], glioblastoma[19], and head and neck cancers[20]. Therefore, the combination of the strong EPR effect of the specific cancers with the hydrophilicity, controlled size (about 80 nm), and enormous boron content of the core-shell nanoparticles allows us to expect the potential applicability of the boronated diblock copolymers to BNCT.

(Response to Query-4)

According to your comment, we added the explanation about the applicability of the EPR effect in humans by citing the relevant literatures, as is follows.

Line 63–70:

However, nowadays, it is emphasized that nanomedicine should focus on more clinically relevant tumor models in terms that the EPR effect of human cancers is extremely heterogeneous in contrast to rodent cancers.[17] Therefore, an appropriate selection of target cancers is of great importance in the nanomedicine based on the passive drug-delivery concept. More specifically, some human cancers exerting a clinically significant EPR effect, such as sarcoma[18], glioblastoma (brain tumor)[19], and head and neck cancers[20], would be promising candidates in nanomedicine-based cancer therapy including BNCT.

 (Response to Query-5)

The aromatic 1H signal of PBA will be overlapped with that of the chain transfer agent CDB in comparable intensity in 1H-NMR spectra (Figure 2S and Figure 2), because aromatic protons of CDB (0.075 mmol) used in the preparation of the copolymer block reaches ca. 0.17 times as much as those of protected PBA (1.13 mmol). According to your advice, therefore, we added the peak assignment of CTA to the text and figures, which is as follows.

Line 232:

the phenyl groups of PBA and CDB (peaks; 3, f, g)

Line 239:

the phenyl groups of PBA and CDB (peaks; 3, f, g)

Figure 2 and Figure S2:

The phenyl groups originating from CDB in both polymer ends were assigned by using the letter symbols of f and g.

(Response to Query-6)

We greatly appreciate your suggestion about the relation of renal clearance and the molecular weight (MW) of polymers, which is one of important factors to be considered in the development of nanomedicines including BNCT agents. As you have pointed out, in the case that individually dispersed diblock copolymers are administered through intravenous injection, there is a possibility that the diblock copolymers would be eliminated them from the body by rapid renal clearance. However, we suppose that not individual biblock copolymers but the boron-enriched core-shell nanoparticles (which would be formed at human body temperature after intravenous administration of the self-assembled nanomicelles at the 1st LCST) will work as boron carriers by utilizing the strong EPR effect of some specific human cancers. As reported previously, for polysaccharides and globular proteins, a hydrodynamic diameter of 3-6 nm (MW; 5-50 kDa) is associated with rapid clearance from the body by renal filtration and urinary excretion [Ref]. Furthermore, the threshold of renal clearance of nanoparticles is ~5.5 nm. Accordingly, our core-shell nanoparticles with hydrodynamic diameter of about 80 nm would not be excreted from body by renal clearance. In the present study, not so high MW of boronated diblock copolymer is more preferred in order to obtain the nanoparticles with properly controlled size of less than 100 nm suitable for the EPR effect. We hope that the above explanation answers your question.

[Ref] Hak Soo Choi, Wenhao Liu, Preeti Misra, Eiichi Tanaka, John P Zimmer, Binil Itty Ipe, Moungi G Bawendi & John V Frangioni. Renal clearance of quantum dots. Nature Biotechnol. 25, 2007, 1165–1170.

(Response to Query-7)

According to your advice, we carried out the additional experiment to determine the critical micelle concentration (CMC) of the diblock copolymer. The measurement result was shown in Figure S5 (Supplementary Information), and the brief explanation of the CMC determined and the potential use of the polymer solution for BNCT were added to revised manuscript, which is as follows.

In Experimental section,

Line 204–211:

The critical micelle concentration (CMC) of the boron-containing diblock copolymer was determined form the concentration-dependent variation of optical transmittance of the polymer solutions at the fixed wavelength of 400 nm (at which the diblock copolymer has no absorbance). The optical transmittance was measured at 20 °C by using UV-Vis spectrophotometer (V-630, JASCO Co.). The scattering of irradiation light on the micelles formed above CMC is responsible for unambiguous decrease of optical transmittance. Therefore, CMC can be determined as the cross point of the two least squares regression lines of the data before and after micellization of the diblock copolymer.

In Results and discussion,

Line 355–359:

The CMC determined from the concentration-dependent variation of transmittance by the scattering of irradiation light on the nanomicelles was ca. 2.8 μM (Figure S5). The quite dilute micellization concentration of the boronated diblock copolymer allows adjustable preparation of nanomicelle solution for clinical use in a wide concentration range.

Reviewer 2 Report

Authors presents in their manuscript the synthe synthesis and characterization of novel type of boronic acid-containing PNIPAN block-copolymer. The materials are well characterized and partly good presented. It gives some requirements for improvements:

-          Introduction (line 70-72): Breaking H-bonds are only possible between NH-amide groups and water and not between water and isopropyl groups (line 72-73)!

-          GPC measurements: do the authors use also other conditions for the determination of Mn and Mw? Dispersity seems to be very high according to RAFT-polymerisation!

-          Authors can improve the Table 1 along to molar ratio between initiator and monomers and theoretical calculation of Mn! Moreover authors are even more suited to calculate Mn of pure PNIPAM and block copolymers by 1H NMR (Figure S2). These results shoud be integrated in table 1.

-          Graphs assignment has to be changed in Figure S4!

-          Table 1 and in text: in gives no polydispersity for GPC, only dispersity Ð. Please change this point!

-          Do the authors have an explanation for the lower LCST value in comparison to literature for PNIPAM! This explanation shoud be integrated in the maintext!

Author Response

To Reviewer 2

We deeply appreciate your courtesy to review our manuscript. Your comments are quite helpful for us to improve the quality of our manuscript. We have made our best efforts to answer your questions and to comply with your advice as much as possible. In addition, we have carefully checked out all data, body text and references again. In this process, we have found some mistakes, and corrected the mistakes appropriately (please see below for the details). We sincerely apologize for our careless mistakes. We would like to emphasize that the major context of “Results and discussion” and “Conclusions” is not changed from that of the original manuscript by this amendment. All the amendments are highlighted with green and sky blue makers in the revised manuscript. The green-highlighted amendments correspond to our response to the questions and requests from you and another reviewer, and the sky blue-highlighted ones are the amendments by our own to improve the manuscript.

We would appreciate it if you could spare your precious time for further consideration of our answer below and revised manuscript.

(Response to Qery-1)

According to you suggestion, we revised the explanation about the LCST phenomenon of PNIPAAm, as is shown in an underlined part below.

Line 79–81:

Poly(NIPAAm) undergoes hydrophilic-hydrophobic phase transition below and above the lower critical solution temperature (LCST) of about 32 °C in aqueous solutions due to the thermally induced hydration/dehydration of the hydrophobic groups (e.g., isopropyl groups) and hydrogen-bond making/breaking between the amide groups and water molecules.[30,31]

(Response to Query-2)

As received your indication, we checked out the GPC data including the measurement conditions. In this process, we have noticed wrong selection of GPC data. Therefore, we have revised GPC chromatograms (Figure S3) and Table 1 using the correct data. As you pointed out, dispersity (Ð) values are comparably small in NIPAAm homopolymer and NIPAAm-PBA copolymer, supporting the controllability of RAFT polymerization. We sincerely apologize for our careless mistakes. In accordance with the corrected Table 1, the explanation about dispersity in the RAFT polymerization of poly(NIPAAm) and poly(NIPAAm-co-PBA) was revised as follows.

Line 278–280:

The Ð values of the NIPAAm-based polymers suggests that the controllability of RAFT polymerization of NIPAAm-copolymer blocks is comparable to that of NIPAAm-homopolymer blocks.

 (Response to Query-3)

According to your advice, the theoretical molecular weight including the molar ratios of chain transfer agent and monomers and the Mn vlues of poly(NIPAAm) and poly(NIPAAm-co-PBA) calculated from Figure S2 were integrated in Table 1 (please see Line 270–273).

 (Response to Query-4)

We deeply apologize our careless mistake.As you pointed out, the assignment in Figure S4 was revised, as is indicated by underline below.

In the caption of Figure S4:

Figure S4. Temperature dependence of optical transmittance of poly(NIPAAm) (open square) and poly(NIPAAm-co-PBA) (open circle) dissolved in pure water.

 (Response to Query-5)

We appreciate your letting us notice the improper term, polydispersity. According to your advice, we replaced polydispersity (PDI) with dispersity (Ð) throughout the manuscript including Table 1, as is follows.

Line188:

molar-mass dispersity (Ð = Mw/Mn)

Line 270–273:

The results of GPC analysis, i.e., the Mw, Mn, and Ð values of poly(NIPAAm), poly(NIPAAm-co-PBA), and poly(NIPAAm-block-NIPAAm-co-PBA) and those values calculated from 1H NMR spectra (Figure S2) are listed in Table 1, which also includes the molar ratios of reaction mixture in RAFT polymerization and theoretical MW.

 (Response to Query-6)

The explanation about the lower LCST of the copolymer block than poly(NIPAAm) was made as follows.

Line 291–293:

The phenyl group of PBA makes poly(NIPAAm-co-PBA) less hydrophilic than poly(NIPAAm), which lowers the LCST of the copolymer compared to poly(NIPAAm).

(Other minor changes)

We checked out the text and figures throughout the manuscript to correct some mistakes including typographical errors. We would like you to see the parts highlighted by blue marker for the amendment. As already mentioned, we noticed wrong selection of GPC data, and hence revised Table 1 using the correct data. In accordance with the revised Table 1, the boron content of the diblock copolymer was re-calculated by using the corrected molecular weight of the diblock copolymer, and changed from 2.1 to 2.9. We sincerely apologize for our careless mistakes.

According to the amendment, some additional literatures were cited, and all of the reference No. were appropriately changed in the revised manuscript.

Round 2

Reviewer 1 Report

The authors have answered all the comments and provided all the required additional info. I think that the manuscript should be accepted in Polymers as is.